
# An improved method of Newmark analysis for mapping hazards of
# coseismic landslides
**Mingdong Zang** [a,b,c], **Shengwen Qi** [a,b,c,*], **Yu Zou** [a,b,c], **Zhuping Sheng** [d], **Blanca S. Zamora** [d]
[a] *Key Laboratory of Shale Gas and Geoengineering, Institute of Geology and Geophysics, Chinese*
*Academy of Sciences, Beijing 100029, China*
[b] *Institutions of Earth Science, Chinese Academy of Sciences, Beijing 100029, China*
[c] *University of Chinese Academy of Sciences, Beijing 100029, China*
[d] *Texas A&M AgriLife Research Center at El Paso, El Paso, Texas 79927, USA*
* Corresponding Author.
*E-mail address:* qishengwen@mail.iggcas.ac.cn (S. W. Qi)



## Abstract

Coseismic landslides have been responsible for destroyed buildings and structures, dislocated roads and bridges, cut off of pipelines and lifelines, and tens of thousands of deaths. Accurately mapping the hazards of coseismic landslides is an important and challenge work. Newmark's method is widely applied to assess the permanent displacement along a potential slide surface to determine the coseismic responses of the slope. This paper considers the roughness and size effect of the potential slide surface-unloading joint, and then presents an improved method of Newmark analysis for mapping hazard of coseismic landslides. The improved method is verified using data from a case study of the 2014 $M_w$ 6.1 (USGS) Ludian earthquake in Yunnan Province, China. The permanent displacement yielded from this method range from 0 to 122 cm. Comparisons are made between the predicted displacements and a comprehensive inventory of landslides triggered by the Ludian earthquake to map the spatial variability using certainty factor model (CFM). Confidence levels of coseismic landslides indicated by certainty factors range from -1 to 0.95. A coseismic landslide hazard map is then produced based on the spatial distribution of the values of certainty factors. Area under the curve analysis is used to draw a comparison between the improved and conventional method of Newmark analysis, revealing the improved performance of the method presented in this paper. Such method can be applied to predict the hazard zone of the region and provide guidelines for making decisions regarding infrastructure development and post-earthquake reconstruction.

*Keywords:* Coseismic landslide; Newmark's method; Barton model; Certainty factor; Hazard Mapping



## 1. Introduction

One of the major causes of landslides is recognized as the earthquake. Coseismic landslide hazards
have drawn increasing attention in recent years (i.e. Jibson et al., 1998, 2000; Khazai and Sitar, 2004; Qi
et al., 2010, 2011, 2012; Chen et al., 2012; Xu et al., 2013; Yuan et al., 2014). In fact, the damage caused
by seismically triggered landslides is sometimes more severe than the damage direct from the earthquake
(Keefer, 1984). Estimating where is likely to have slope failure under a specific shaking condition plays
an important role in regional assessment of coseismic landslides.
Pseudostatic analysis formalized by Terzhagi (1950) and finite-element modeling applied by Clough
and Chopra (1966) were employed to assess the seismic stability of slopes in early efforts (Jibson, 2011).
Newmark (1965) first introduced a relatively simple and practical method, still commonly used, to
estimate the coseismic permanent-displacements of slopes (Jibson, 2011). Studies showed that
Newmark's method yields reasonable and practical results when modeling the dynamic performance of
natural slopes (Wilson and Keefer, 1983; Wieczorek et al., 1985; Jibson et al., 1998, 2000; Pradel et al.,
2005). Recent years, Rathje and Antonakos (2011) present a unified framework for predicting coseismic
permanent sliding displacement based on Newmark's method. Chen et al. (2018) used Newmark's method
to calculate the minimum accelerations required for coseismic landslides in the affected region of 2014
Ludian earthquake. Chen et al. (2019) subsequently developed an easy-operation mapping method to
assess coseismic landslide hazard in the quake zone of 2014 Ludian earthquake, with the help of
Newmark's method.
Such applications generally start from an analysis of the dynamic stability of slopes that is quantified
as the critical acceleration. Barton model (Barton, 1973) has been widely used in rock mechanics and



engineering field to predict the shear strength of rock joints, which plays a crucial role in the calculation
of critical acceleration. However, researches do not pay enough attentions on the shear strength of rock
joints during the assessment of coseismic landslides. To better estimate the dynamic stability of slopes, in
this paper, we introduce the Barton model (Barton, 1973) into a Newmark analysis to develop an improved
modeling method for mapping hazards of coseismic landslides, using data from the 2014 Ludian
earthquake in Yunnan Province, Southwestern China. As predictions of coseismic landslides are not only
based on exact results, i.e., computed permanent-displacements, but also mingled with unformalized
expertise (Shortliffe and Buchanan, 1975), i.e., interpreted landslides, we then present a model of inexact
reasoning method, which defies analysis as applications of sets of inference rules that are expressed in
the predicate logic (Shortliffe and Buchanan, 1975), to produce a coseismic landslide hazard map.
This paper briefly introduces the site characteristics and the spatial distribution of triggered
landslides, describes the modeling method used for the analysis of seismic slope stability, then presents
the mapping procedure of the confidence level of seismic slope-failure, and finally discusses the results
of the seismic hazard assessment and the comparison with a conventional Newmark analysis.

**2. Study area**
The epicenter of the 2014 $M_w$ 6.1 Ludian earthquake is located in the southeastern margin of the
Tibetan plateau. A rectangular area lying immediately around the epicenter and containing dense
concentrations of induced landslides was chosen for study. Elevation in the study area ranges from 785
m to 3,085 m above the sea. There are three rivers, the Niulanjiang River, the Shaba River and the
Longquan River passing through the area. The topography ranges from flat in river valleys to nearly





vertical in the slopes on the side of the rivers. The Niulanjiang River, flowing from southeast (SE) to the
northwest (NW), where according to Chen et al. (2015), incises down to a depth between 1,200 m and
3,300 m, resulting in about 80% of the slopes with angles greater than 40° distributed along the banks.
Predominant geologic units of the study area vary in the era from Proterozoic to Mesozoic, including
dolomite, limestone, shale, sandstone, basalt and slate.
A landslide inventory containing 1,416 landslides (Fig. 1) was posed by visual interpretation through
comparison between pre-earthquake satellite images from Google Earth (January 30, 2014) and 0.2m-
high-resolution post-earthquake aerial images (August 7, 2014, data provided by Digital Mountain and
Remote Sensing Applications Center, Institute of Mountain Hazards and Environment, Chinese Academy
of Sciences; Beijing Anxiang Power Technology Co., LTD.). A majority of landslides triggered in this
earthquake were shallow flow-like landslides (less than 3 m deep) developing in particularly dense
concentrations along steeply incised river valleys. The total area of these interpreted landslides was 7.01
km$^2$ within a study area of 705 km$^2$. A detailed study showed that 846 of the mapped landslides were
greater than 1,000 m$^2$, occupying 6.74 km$^2$ and accounting for 96.1% of the total landslide area, out of
which 279 of the mapped landslides were greater than 5,000 m$^2$, occupying 5.37 km$^2$ and accounting for
76.6% of the total landslide area.

**3. Methodology**
3.1 Modeling method
In the context of the analysis of the dynamic stability of a slope, Newmark (1965) proposed a
permanent-displacement analysis that bridges the gap between simplistic pseudostatic analysis and



sophisticated, but generally impractical finite-element modeling (Jibson, 1993). Newmark's method
simulates a landslide as a rigid-plastic friction block having a known critical acceleration on an inclined
plane (Fig. 2), and then calculates the cumulative permanent displacement of the block as it is subjected
to an acceleration-time history of an earthquake. Newmark (1965) showed that the dynamic stability of a
slope is related to the critical acceleration of a potential landslide block, and it can be expressed as a
simple function of the static factor of safety and the landslide geometry (Jibson et al., 1998, 2000) as
below:

$$a_c = (F_S - 1)g\sin\alpha \tag{1}$$

where $a_c$ is critical acceleration in terms of $g$, the acceleration due to earth's gravity, $F_S$ is static factor
of safety, and $\alpha$ is the angle from the horizontal that the center of the slide block moves when
displacement first occurs (Jibson et al., 1998, 2000). For a planar slip surface parallel to the slope, this
angle can generally be approximated as the slope angle.
Natural slopes often develop a group of shallow unloading joints (Fig. 3) that parallel to the surface
due to valley incisions (Gu, 1979; Hoek and Bray, 1981). Studies showed that rock slopes behave as
collapsing and sliding failures of shallow unloading joints under strong earthquakes, and 90% of
coseismic landslides are shallow falls and slides (Harp and Jibson, 1996; Khazai and Sitar, 2003; Dai et
al., 2011; Tang et al., 2015). According to Qi et al. (2012), there are two typical kinds of earthquake
triggered landslides, i.e., (a) shallow flow-like landslides with depth less than 3 m in general and (b) rock
falls that are thrown by the earthquake shaking, usually occurred at the crest of the slope. For both types,
the unstable rock blocks are often cut and activated along the rock joints. Therefore, the static factor of
safety in terms of the critical acceleration in these conditions is related to the peak shear strength of the


rock joints. For the purpose of regional stable analysis, we use a limit-equilibrium model of an infinite
slope (Fig. 2) referring to the simplification of Jibson et al. (1998, 2000) on Newmark's method. On this
occasion, the value of the static factor of safety against sliding which is given by the ratio of resisting to
driving force is determined by conventional analysis with no consideration of accelerations, expressed as:

$$F_S = \frac{Resisting\ force}{Driving\ force} = \frac{\tau L}{mg sin\alpha} = \frac{\tau L}{\gamma L t sin\alpha} = \frac{\tau}{\gamma t sin\alpha} \tag{2}$$

where $\tau$ is peak shear strength of the rock joint, $\gamma$ is unit weight of the rock mass, and $t$ is the thickness
of the failure rock block.
For a Newmark analysis, it has been customary to describe the shear strength of rocks not rock joints
in terms of Coulomb's constants for friction and cohesion. However, both are not only stress dependent
variables, but also scale dependent (Barton and Choubey, 1977). According to Barton (1973), a more
satisfactory empirical relationship for predicting the peak shear strength of a joint can be written as
follows:

$$\tau = \sigma_n \tan\left[JRC\ log_{10}\left(\frac{JCS}{\sigma_n}\right) + \phi_b\right] \tag{3}$$

where $\sigma_n$ is effective normal stress, $JRC$ is joint roughness coefficient, $JCS$ is joint wall compressive
strength, $\phi_b$ is basic friction angle, the angle of frictional sliding resistance between rock joints, which
can be obtained from residual shear tests on natural joints (Barton, 1973).
The effective normal stress ($\sigma_n$) generated by the gravity acting on the rock block is as follows:

$$\sigma_n = \frac{mg cos\alpha}{L} = \frac{\gamma L t cos\alpha}{L} = \gamma t cos\alpha \tag{4}$$



Considering the impact of size effect on $JRC$ and $JCS$, formulations were developed by Barton and
Bandis (1982) and are shown as below:

$$JRC_n = JRC_0 \left(\frac{L_n}{L_0}\right)^{-0.02JRC_0} \tag{5}$$

$$JCS_n = JCS_0 \left(\frac{L_n}{L_0}\right)^{-0.03JRC_0} \tag{6}$$

where the nomenclature adopted incorporates the (*0*) and (*n*) for laboratory scale and in situ scale values
respectively.
Hence the static factor of safety ($F_S$) of a slope can be written as:

$$F_S = \frac{\tau}{\gamma t sin\alpha} = \frac{\sigma_n tan\left[JRC_n\, log_{10}\left(\frac{JCS_n}{\sigma_n}\right) + \phi_b\right]}{\gamma t sin\alpha}$$

$$= \frac{\gamma t cos\alpha\, tan\left[JRC_n\, log_{10}\left(\frac{JCS_n}{\gamma t cos\alpha}\right) + \phi_b\right]}{\gamma t sin\alpha}$$

$$= \frac{tan\left[JRC_n\, log_{10}\left(\frac{JCS_n}{\gamma t cos\alpha}\right) + \phi_b\right]}{tan\alpha} \tag{7}$$

After knowing the slope angle and the static factor of safety, the critical acceleration of a slope can
be determined. Once the earthquake acceleration-time history has been selected, those portions of the
record lying above the critical acceleration $a_c$ (Fig. 4a) are integrated once to derive a velocity profile
(Fig. 4b), which in turn is integrated a second time to obtain the cumulative displacement profile of the
block (Fig. 4c), users then judge the dynamic performance of a slope based on the magnitude of the



Newmark displacement (Jibson et al., 1998, 2000; Jibson, 2011). The detailed procedure of conducting a
Newmark analysis with Barton model is discussed in the following sections.
3.2 Static factor of safety
Considering that the mapped landslides greater than 1,000 m$^2$ occupy 96.1% of the total landslide area,
we selected a 30 m×30 m digital elevation model (DEM), ASTER Global Digital Elevation Model
(https://doi.org/10.5067/ASTER/ASTGTM.002, last accessed July 16, 2018) that is capable of facilitating
the subsequent hazard analysis. A basic slope algorithm was applied to the DEM to produce a slope map
(Fig. 5), where the slope is identified as the steepest downhill descent from the cell to its neighbors
(Burrough and McDonell, 1998). The slopes range from greater than 60° in the banks of the Niulanjiang
River, the Shaba River and the Longquan River, to less than 20° in moderate and low mountains and hills
in north and east.
For some slope steeper than 60°, few blocks can stay on that steep sliding surface, and the calculated
$F_S$ will be nearly zero in this case. Actually, the unstable blocks have already failed, and further sliding
will occur along a failure plane inside the slope, and the angle (α) of the inclination of the failure plane
will be $45°+\frac{\phi_b}{2}$. Therefore, we assigned an angle (α) of $45°+\frac{\phi_b}{2}$ to those slopes more than 60° to avoid a
too low $F_S$ in Newmark analysis.
Digital geologic map from China Geological Survey (GCS) was rasterized at 30 m grid spacing for
assigning material properties throughout the study area. According to the literature researches, we found
that $JRC_0$ and $JCS_0$ depend strongly on the lithology (Coulson, 1972; Barton and Choubey, 1977;
Bandis et al., 1983; Priest, 1993; Bilgin and Pasamehmetoglu, 1990; Singh et al., 2012 Alejano et al.,
2012, 2014; Giusepone, 2014; Yong et al., 2018). Representative values of $\gamma$, $JRC_0$, $JCS_0$ and $\phi_b$


assigned to each rock type exposed in the area can normally be estimated with the help of the test data
listed in Table 1. The selected values were near the middle of the ranges represented in the references.
These $JRC_0$ and $JCS_0$ are considered in laboratory scale, for the length of 100 mm as $L_0$. For each grid
cell in regional analysis, $L_n$, the length of engineering dimension, can generally be set as a ten-fold range
of $L_0$, because the value of $JRC_n/JRC_0$ ($JCS_n/JCS_0$) is almost constant when the value of $L_n/L_0$
greater than 10 (Bandis et al., 1981). The values of $JRC_n$ and $JCS_n$, then, are calculated by inserting
values from $JRC_0$, $JCS_0$, $L_0$, and $L_n$ into Eq. (5) and Eq. (6). Fig. 6a and Fig. 6b show the spatial
distribution of $JRC_n$ and $JCS_n$ respectively. The basic-friction-angle ($\phi_b$) map and unit weight ($\gamma$) map
are shown as Fig. 7 and Fig. 8 respectively.
For simplicity, the thickness of the modeled block $t$ was taken to be 3 m, which reflects the typical
slope failures of the Ludian earthquake. The static factor-of-safety map was produced by combing these
data layers ($\alpha$, $JRC_n$, $JCS_n$, $\phi_b$, and $\gamma$) in Eq. (7). In the initial iteration of the calculation, grid cells in
steep areas with static factors of safety less than 1 indicate that the slopes are statically unstable, but do
not necessarily mean that the slopes are moving under the earthquake shaking. In this condition, to avoid
conservative results, we did not increase the strengths of rock types having statically unstable cells, either,
adjust strengths of other rock types to preserve the relative strength differences between rock types (Jibson
et al., 1998, 2000). Instead we assigned a minimal static factor of safety as 1.01, merely above limit
equilibrium (Jibson et al., 1998, 2000), to these slopes, to avoid a negative value of the critical acceleration
$a_c$. According to Keefer (1984), most landslides triggered by earthquakes occur with a slope of 5° at least.
Static factors of safety resulting from slopes less than 5° were very high, and these slopes that were
impossible to have failures under the Ludian earthquake did not produce a statistically significant sample





to the analysis. Therefore, slopes less than 5° were not analyzed during the second iteration. After the
adjustment, the static factors of safety ranged from 1.0 to 17.4, as shown in Fig. 9.
3.3 Critical acceleration
According to Newmark (1965), a pseudostatic analysis in terms of the static factor of safety and the
slope angle was employed to calculate the critical acceleration of a potential landslide. The critical-
acceleration map (Fig. 10) was produced by combining the static factor of safety and the slope angle in
Eq. (1).
The critical acceleration that results in a static factor of safety of 1.0 and initiates a sliding of a slope in
a limit-equilibrium analysis is derived from the intrinsic slope properties (topography and lithology),
regardless which ground shaking is given. Therefore, the critical-acceleration map indicates the
susceptibility of the coseismic landslides (Jibson et al., 1998, 2000). The calculated critical accelerations
range from almost zero in areas that are more susceptible to coseismic landslides, to 14.0 $g$ in areas with
lower susceptibility.
3.4 Shake map
There are 23 strong-motion stations within 100 km of the Ludian earthquake epicenter (Fig. 11).
Each station record includes three components of the peak ground acceleration ($PGA$), in south-north
direction, east-west direction and up-down direction respectively, as listed in Table 2 (The data set is
provided by China Earthquake Data Center, http://data.earthquake.cn, last accessed June 16, 2016). We
calculated the average $PGA$ of the two horizontal components of each strong-motion recording, and then
plotted a contour map (Fig. 12) using an Inverse Distance Weighted (IDW) interpolation algorithm. This
method assumes that the variable of the average $PGA$ being mapped decreases in influence with distance




from its sampled location. Inverse Distance Weighted (IDW) interpolation determines cell values using a
linearly weighted combination of a set of sample stations (Watson and Philip, 1985). The weight is a
function of inverse distance. In addition, considering that input stations far away from the epicenter
location where the prediction is being made may have poor or no spatial correlation, we eliminated the
input stations out of 100 km from the calculation.
3.5 Newmark displacement
In a real landslide hazard case, it is impossible to conduct a rigorous Newmark analysis when
accelerometer records are unavailable. It is also impractical and time consuming to produce a
displacement in each cell during the regional analysis. Therefore, empirical regressions (Ambraseys and
Menu, 1988; Bray and Travasarou, 2007; Jibson, 2007; Saygili and Rathje, 2008; Rathje and Saygili,
2009; Hsieh and Lee, 2011) were proposed to estimate Newmark displacement as a function of the critical
acceleration and peak ground acceleration or Arias intensity. Among those empirical estimations, Rathje
and Saygili (2009) developed a vector model for displacement in terms of the critical acceleration ($a_c$),
peak ground acceleration ($PGA$) and moment magnitude ($M_w$) based on analysis of over 2,000 strong
motions.

$$lnD = 4.89 - 4.85\left(\frac{a_c}{PGA}\right) - 19.64\left(\frac{a_c}{PGA}\right)^2 + 42.49\left(\frac{a_c}{PGA}\right)^3 - 29.06\left(\frac{a_c}{PGA}\right)^4$$

$$+0.72\ln(PGA) + 0.89(M_w - 6) \tag{8}$$

where $D$ is predicted displacement in units of $cm$, $a_c$ and $PGA$ are in units of $g$.



This model is a preferred displacement model at a specific site where acceleration-time recordings are
not available. The incorporating multiple ground motion parameters in the analysis typically results in
less variability in the prediction of displacement (Rathje and Saygili, 2009).
The Newmark displacement (Fig. 13) in each cell was calculated by combing corresponding values
of the critical acceleration, peak ground acceleration and moment magnitude in Eq (8). Predicted
displacements range from 0 cm to 122 cm.
3.6 Certainty factor and coseismic landslide hazard map
According to Jibson et al. (1998, 2000), predicted displacements provide an index of seismic
performance of slopes, larger predicted displacements relate to greater incidence of slope failures. But the
displacements do not correspond directly to measurable slope movements in the field. To produce a
coseismic landslide hazard map, we chose a model of inexact reasoning, the certainty factor model (CFM),
which was created by Shortliffe and Buchanan (1975) and improved by Hecherman (1986), to explore
the relationship between the landslide occurrences and the predicted displacements. The CFM was created
as a numerical method, which was initially used by MYCIN, a backward chaining expert system in
medicine (Shortliffe and Buchanan, 1975), for managing uncertainty in a rule-based system. In this model,
the certainty factor $CF$ represents the net confidence in a hypothesis $H$ based on the evidence $E$
(Hecherman, 1986). Certainty factors range between -1 and 1. A $CF$ with a value of -1 means total lack
of confidence, whereas a $CF$ with a value of 1 means total confidence. Values greater than 0 favor the
hypothesis while values less than 0 favor the negation of the hypothesis. According to Hecherman (1986),
there is a probabilistic interpretation for $CF$ shown as below:




$$CF = \begin{cases} \dfrac{p(H|E) - p(H)}{p(H|E)[1 - p(H)]}, & p(H|E) > p(H) \\[4mm] \dfrac{p(H|E) - p(H)}{p(H)[1 - p(H|E)]}, & p(H|E) < p(H) \end{cases} \tag{9}$$

where $CF$ is the certainty factor, $p(H|E)$ denotes the conditional probability for the case of a posterior
hypothesis that relies on evidence, the posterior probability, and $p(H)$ is the prior probability before any
evidence is known. In the displacement analysis, $p(H|E)$ was defined as the proportion of the landslide
area within a specific displacement area while $p(H)$ was defined as the proportion of the landslide area
within the entire study area excluding the slopes less than 5°. In this way, values of $CF$ represent the
confidence level of coseismic landslides. Positive values correspond to an increase in confidence level in
a slope failure while negative quantities correspond to a decrease in confidence level. Greater positive
values indicate higher confidence level of coseismic landslides.
Given this definition, we could produce a coseismic landslide hazard map in terms of certainty factors.
First, displacement cells in every 1 cm were grouped into bins, such that all cells having displacements
between 0 cm and 1 cm were grouped into the first bin; those having displacements between 1 cm and 2
cm were grouped into the second bin, and so on. The displacements were grouped into 123 bins, from 0
cm to 122 cm. Later, we calculated the proportion of cells occupied by landslide area in each bin. This
proportion was considered the posterior probability of each bin as defined. The prior probability
calculated by dividing the entire landslide area by the entire study area is same in each bin. Finally, values
of $CF$ were computed in each bin by using Eq. (9) to combine corresponding values of the posterior
probability and prior probability. Certainty factors range from -1 to 0.95. Values of $CF$ indicate the



confidence level of landslide occurrence of each bin in the study area and provide the basis for producing
a coseismic landslide hazard map.
As shown in the hazard map for the Ludian earthquake (Fig. 14), most of the actual triggered landslides
lie in the higher confidence-level areas with $CF$ values greater than 0.60. The interpreted landslides are
covered on the map to demonstrate the good fit for predicted confidence levels of coseismic landslides.

**4. Results and Discussion**
The predicted displacements represent the cumulative sliding displacements for a given
acceleration-time history. Based on the statistically significant sizes of the area of each displacement,
displacements less than 60 cm, which is around the middle of the displacement range, occupy about 80%
of the study area, while displacements greater than 80 cm occupy a very small area. Jibson et al. (1998,
2000) supposed that shallow falls and slides in brittle, weakly cemented materials would fail at a relatively
small displacement, while slumps and block slides in more compliant materials would likely fail at a
larger displacement. That is to say, the study area is more susceptible to rock falls and shallow, disrupted
slides that fail at a relatively small displacement, while the study area is with a lower probability subjected
to coherent, deep-seated slides that would fail at a larger displacement. Indeed, the majority of landslides
triggered by the Ludian earthquake were shallow, disrupted slides and rock falls (Zhou et al., 2016).
Although few catastrophic rock avalanches, such as the Hongshiyan landslide (Chang et al., 2017),
occurred in the field, they did not produce statistically significant samples that could meaningfully
contribute to the model, which was consistent with the statistic results as discussed previously. Therefore,



the model should relate well to typical kinds of earthquake-induced landslides in the study area,
meanwhile demonstrate its potential utility to predict the probability of other types of landslides.

For each $CF$-value area, the proportion of area occupied by landslide area was plotted as a dot in

Fig. 15. The data was fitted by a piecewise function, which was derived from Eq. (9). Different from a
Weibull curve (1939) through statistical regression, whose shape would probably be different in different
regions (Jibson et al., 1998, 2000), the piecewise function of $CF$ value and the proportion of landslide
area can be derived from Eq. (9). This method is more universal. From the curve shown in Fig. 15, when
the value of $CF$ is reaching 1.0 (total confidence), the proportion of landslide area is trending to
monotonically increase, which means the confidence level of a slope failure is growing and a landslide
would probably occur. Such a procedure is consistent with the interpretation of the certainty factor theory.
Therefore, the CFM demonstrates the capability of its representation and predicting approach for a
probabilistic hazard analysis of coseismic landslides.
When fitting the results of shear tests using Coulomb's linear relation, the shear strengths vary widely
from high normal stress in laboratory to low normal stress in the field (Barton, 1973). We introduced
Barton model into the Newmark analysis to reduce the variability of shear strengths in terms of Coulomb's
constants. And we considered the impact of scale effects by using Eq. (5) and Eq. (6), which helps to
prevent Newmark's method from underestimating the shear strength of geologic units in a regional
analysis. In addition, for Barton model, the joint roughness coefficient ($JRC$) could be estimated from tilt
tests or from matching of Barton joint standard roughness profiles that were regarded by the International
Society for Rock Mechanics (ISRM, 1978), while the joint wall compressive strength ($JCS$) could be
estimated by Schmidt hammer index tests. These tests are helpful to make a quick estimate of the shear





strength in situ, which could facilitate using Newmark's method in an emergency hazard and risk
assessment after an earthquake.

It is difficult for a statically stable slope to fail under an earthquake. Earthquakes usually make statically

unstable slopes or slopes on the boundary fail. For this reason, it is important to truthfully characterize
the shear strengths of slopes. Shear strengths assigned to the geologic units were from results of hundreds
of shear tests from the references. We assigned the original shear strengths to the geologic units other than
increasing strengths to make statically unstable cells stable as Jibson et al. (1998, 200) did, which will
change the statically stable level of the whole area, especially the slopes on the boundary at first. In
addition, we considered size effect of the potential slide surface, this would yield lower $F_S$, which, in turn,
yield higher displacement. However, the actual inventory of landslides was used to calibrate the predicted
displacements, and the confidence levels indicated by certainty factors fit well of the spatial distribution
of coseismic landslides as shown in the hazard map (Fig. 14).

We also ran a conventional Newmark analysis using assigned strengths, such as internal friction angle

($\varphi$) and cohesion ($c$) as shown in Table 2. Predicted displacements calculated by the conventional
Newmark analysis range from 0 cm to 121 cm, compared with 0 cm to 122 cm by the new method
described in the paper. Fig. 16 shows the hazard map produced using the conventional Newmark analysis.
The $CF$s range from -1 to 0.94, almost the same as results from the new method above. However, there
are big differences along the Shaba River and upstream of the Niulanjiang River from these two methods.
By comparing Fig. 14 with Fig. 16, we can see that confidence levels from the new method fit better than
that of the conventional method, especially near upstream of the Niulanjiang River. The area under the
curve (AUC) analysis was employed to compare performances of both methods. To create an AUC plot,


the cumulative area of $CF$s within each interval of calculated values from the maximum to the minimum
was determined as a proportion of the total study area (x-axis) and plotted against the proportion of
cumulative landslides falling within those $CF$s (y-axis) (Miles and Keefer, 2009). The area under the
curve is calculated as an index to conduct comparison across both methods. A value of 0.5 indicates
performance that is no better than random guessing and 1.0 indicates perfect performance (Miles and
Keefer, 2009). Fig. 17 shows the results of the AUC analysis for both methods. The calculated AUC value
for the new method is 0.58, while the value for the conventional Newmark's method is 0.53. That is to
say, the new method introduced in this paper yields better results, and it is actually an improvement over
the conventional way of Newmark analysis.

**5. Conclusion**
Newmark's method is a useful, physically based model to estimate the seismic stability of natural slopes.
Mapping procedure of data from the 2014 Ludian earthquake shows the feasibility of a Newmark analysis
combined with Barton' shear strength criterion. Such method provides practical applications in regional
seismic hazard assessment. We also consider the size effect of shear strength parameters, such as the joint
roughness coefficient ($JRC$) and the joint wall compressive strength ($JCS$) in a regional analysis. Moreover,
the linkage of Newmark displacements to certainty factor model improves the utility of Newmark's
method to predict the hazard of coseismic landslides. Finally, results of the AUC analysis indicate that the
new method has higher reliability than a conventional Newmark's method.

**Acknowledgements**





This work is supported by Natural Science Foundation of China under Grants of Nos. 41825018 and
41672307, Science and Technology Service Network Initiative under Grant No. KFJ-EW-STS-094, and
the sponsorship from the China Scholarship Council (No. 201704910537).



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



**Figure Captions**

**Fig. 1.** Map of the study area showing interpreted landslides.

**Fig. 2.** Conceptual sliding-block model of a Newmark analysis.

**Fig. 3.** A schematic diagram showing shadow unloading joints in the slope.

**Fig. 4.** Demonstration of the Newmark-analysis algorithm (adapted from Wilson and Keefer, 1983; Jibson et al., 1998, 2000)

**Fig. 5.** Slope map derived from the DEM of the study area.

**Fig. 6.** (a) $JRC_n$ component and (b) $JCS_n$ component of shear strength assigned to rock types in the study area.

**Fig. 7.** Basic-friction-angle ($\phi_b$) component of shear strength assigned to rock types in the study area.

**Fig. 8.** Unit weight ($\gamma$) assigned to rock types in the study area.

**Fig. 9.** Static factor-of-safety map of the study area.

**Fig. 10.** Map showing critical accelerations in the study area.

**Fig. 11.** Locations of strong-motion stations.

**Fig. 12.** Contour map of peak ground acceleration ($PGA$) produced by the Ludian earthquake in the study area. $PGA$ values shown are in $g$.

**Fig. 13.** Map showing predicted displacements in the study area.

**Fig. 14.** Map showing confidence levels of coseismic landslides in the Ludian earthquake using method introduced in this paper. Confidence levels are portrayed in terms of values of $CF$.


**Fig. 15.** Proportion of the area of landslides lying in each $CF$-value area. A dot shows the proportion of
landslide area within an area of $CF$ value; the red line is the fitting curve of the data using second order
exponential growth function.
**Fig. 16.** Map showing confidence levels of coseismic landslides in the Ludian earthquake using a
conventional Newmark analysis. Confidence levels are portrayed in terms of values of $CF$.
**Fig. 17.** Area under the curve plots for comparing the new method with a conventional Newmark's
method.



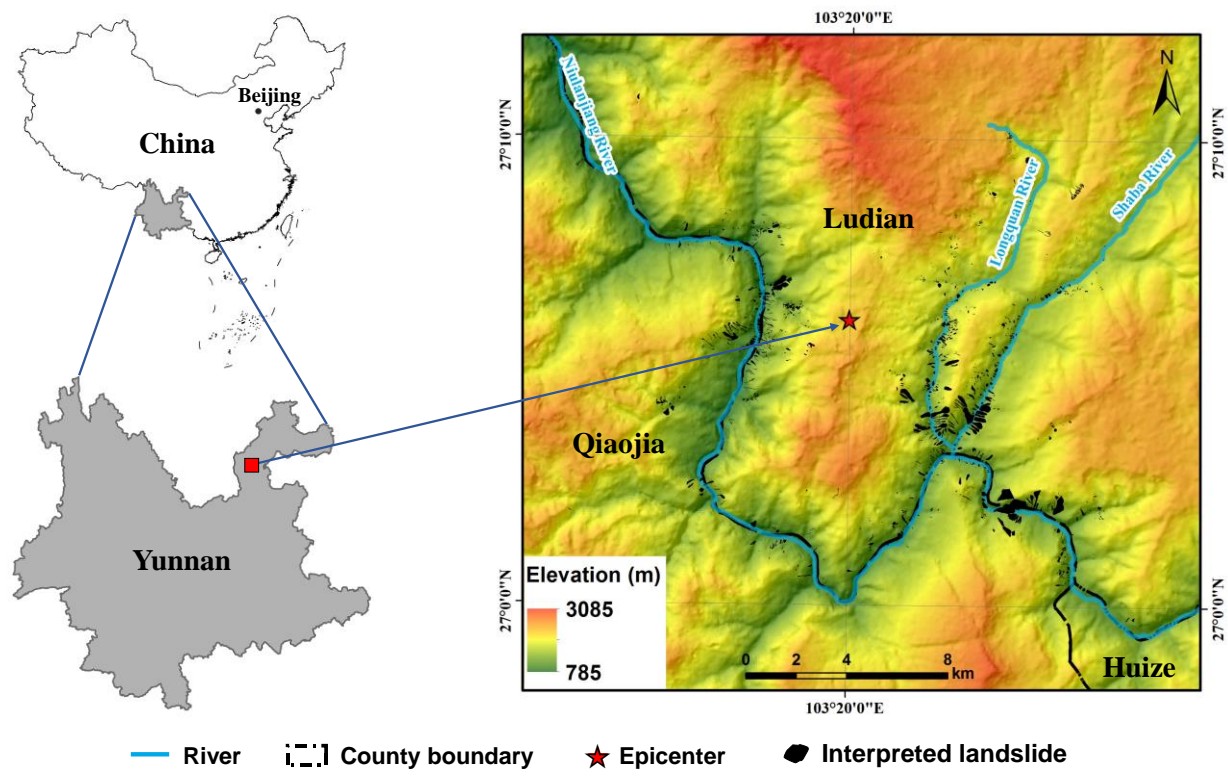


**Fig. 1.** Map of the study area showing interpreted landslides.




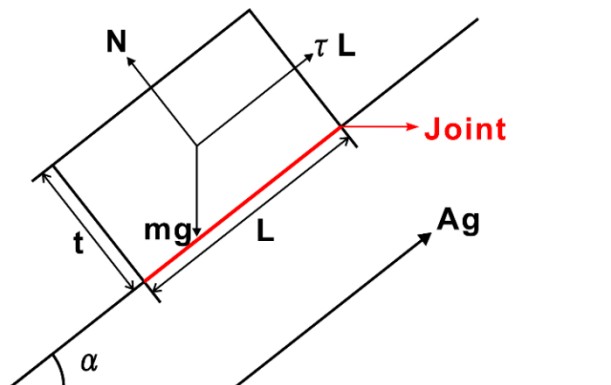


**Fig. 2.** Conceptual sliding-block model of a Newmark analysis. The potential landslide is modeled as a

rigid-plastic block resting on an inclined plane at an angle ($\alpha$) from the horizontal (Jibson et al., 1998,

2000). The base of the block is subjected to an earthquake ground acceleration that is denoted by $Ag$.





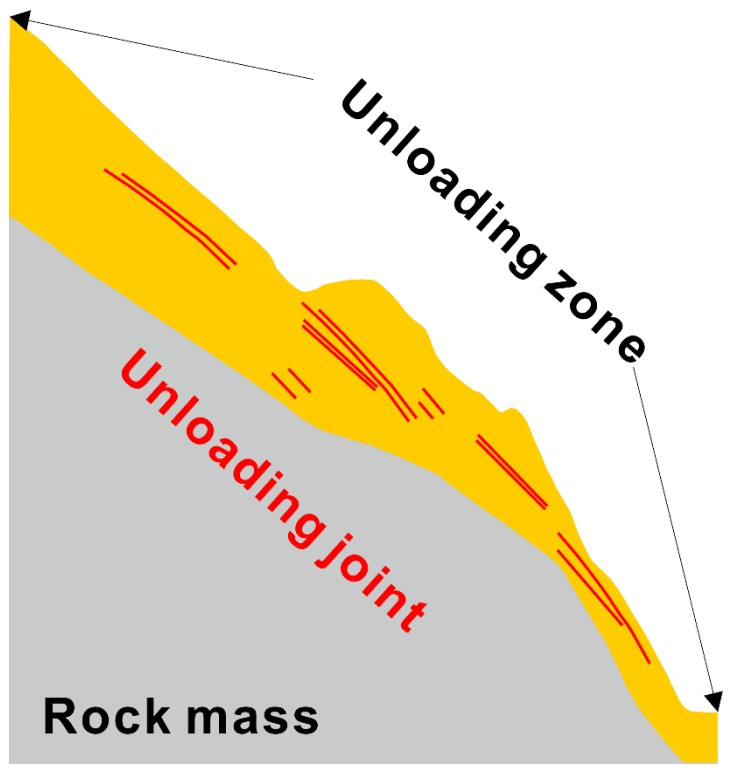


**Fig. 3.** A schematic diagram showing shallow unloading joints in the slope.



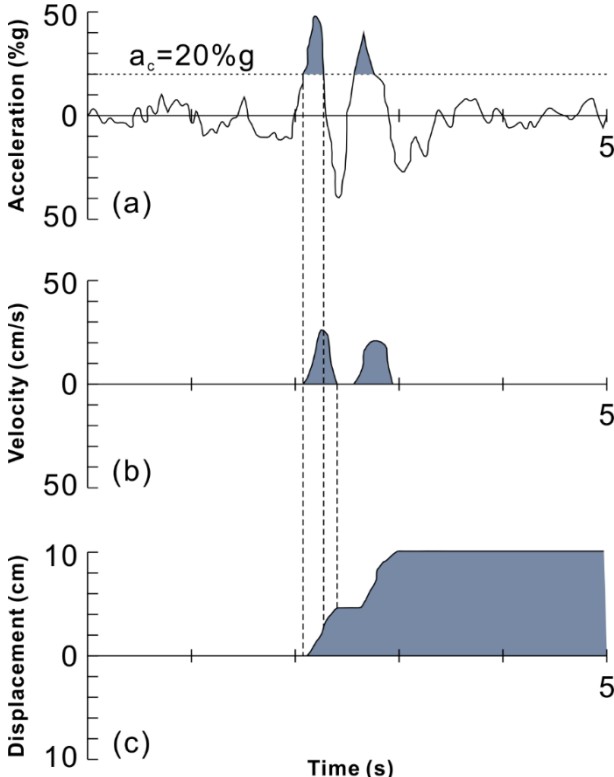


**Fig. 4.** Demonstration of the Newmark-analysis algorithm (adapted from Wilson and Keefer, 1983; Jibson

et al., 1998, 2000): (a) Acceleration-time history with critical acceleration (horizontal dotted line) of 20%g

superimposed. (b) Velocity of block versus time. (c) Displacement of block versus time.




**Fig. 5.** Slope map derived from the DEM of the study area.



(a)





(b)
**Fig. 6.** (a) $JRC_n$ component and (b) $JCS_n$ component of shear strength assigned to rock types in the
study area.
**Fig. 7.** Basic-friction-angle ($\phi_b$) component of shear strength assigned to rock types in the study area.





**Fig. 8.** Unit weight ($\gamma$) assigned to rock types in the study area.



**Static factor of safety**

1.0  1.1  1.5  2.0  2.5  3.0  3.5  5.5  6.5    Slopes < 5°


**Fig. 9.** Static factor-of-safety map of the study area.

**Fig. 10.** Map showing critical accelerations in the study area.






**Fig. 11.** Locations of strong-motion stations.



**Fig. 12.** Contour map of peak ground acceleration ($PGA$) produced by the Ludian earthquake in the

study area. $PGA$ values shown are in $g$.




**Fig. 13.** Map showing predicted displacements in the study area.


**Fig. 14.** Map showing confidence levels of coseismic landslides in the Ludian earthquake using method

introduced in this paper. Confidence levels are portrayed in terms of values of *CF*.


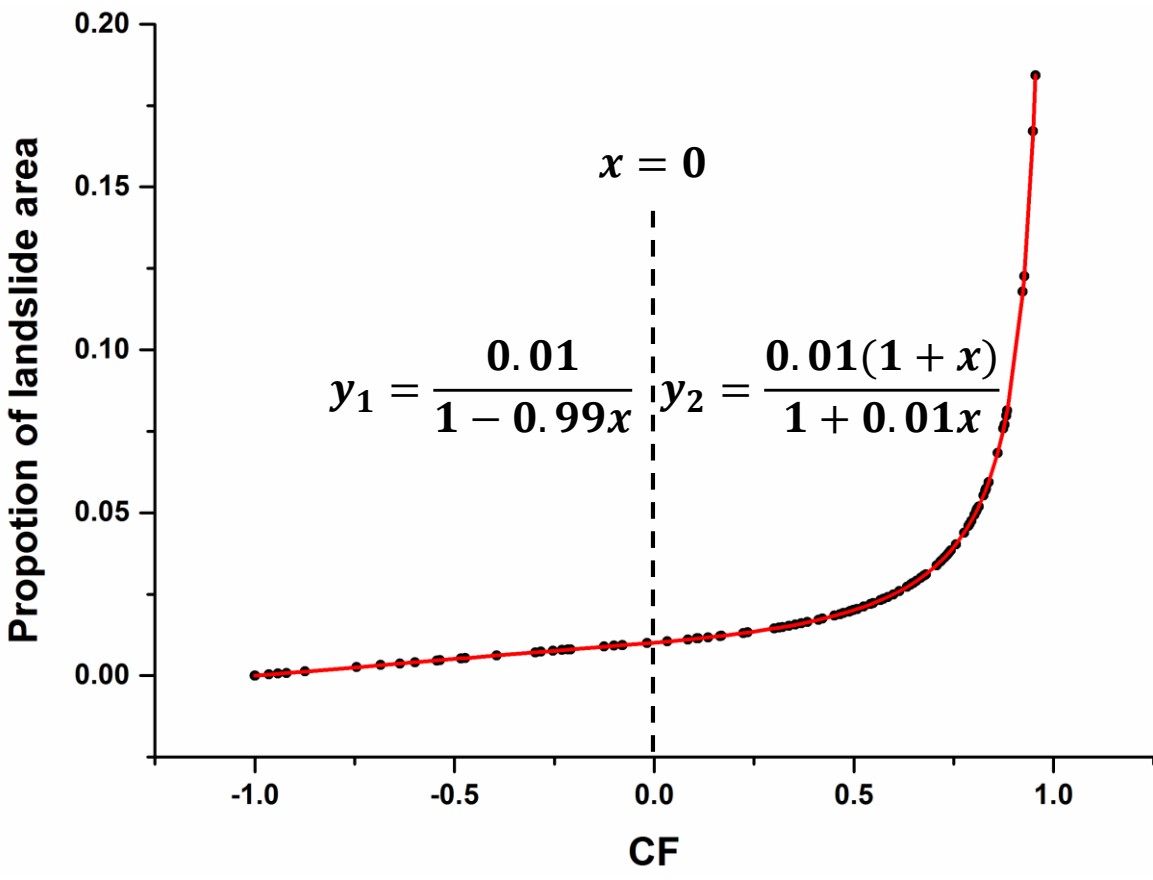

558

**Fig. 15.** Proportion of the area of landslides lying in each $CF$-value area. A dot shows the proportion of

landslide area within an area of $CF$ value; the red line is the fitting curve of the data using second order

exponential growth function.

562



**Fig. 16.** Map showing confidence levels of coseismic landslides in the Ludian earthquake using a conventional Newmark analysis. Confidence levels are portrayed in terms of values of *CF*.








**Fig. 17.** Area under the curve plots for comparing the new method with a conventional Newmark's

method.




**Table Captions**
**Table 1.** Shear strengths assigned to rock types in the study area.
**Table 2.** Station records included three components of the peak ground acceleration.



**Table 1**
Shear strengths assigned to rock types in the study area.

| Rock type | $\gamma$ (kN/m³) | $\phi_b$ | $JCS_0$ (MPa) | $JRC_0$ | $\varphi$ | $c$ (kPa) | References |
|---|---|---|---|---|---|---|---|
| Dolomite | 25.9 | 32° | 140 | 9.5 | 43° | 35 | Singh et al., 2012 |
| | | | | | | | Giusepone, 2014 |
| | | | | | | | Alejano et al., 2014 |
| | | | | | | | Bandis et al., 1983 |
| Limestone | 21.5 | 37° | 160 | 9 | 45° | 30 | Singh et al., 2012 |
| | | | | | | | Yong et al., 2018 |
| | | | | | | | Barton and Choubey, 1977 |
| Shale | 24.9 | 27° | 75 | 8 | 27° | 16 | Bilgin and Pasamehmetoglu, 1990 |
| | | | | | | | Coulson, 1972 |
| Sandstone | 23.5 | 35° | 100 | 6 | 42° | 24 | Bandis et al., 1983 |
| | | | | | | | Priest, 1993 |
| | | | | | | | Coulson, 1972 |
| Basalt | 27.9 | 38° | 205 | 8.5 | 50° | 40 | Barton and Choubey, 1977 |
| | | | | | | | Alejano et al., 2014 |
| | | | | | | | Coulson, 1972 |
| | | | | | | | Barton and Choubey, 1977 |
| Slate | 26.5 | 30° | 175 | 3 | 40° | 11 | Bandis et al., 1983 |
| | | | | | | | Alejano et al., 2012 |
| | | | | | | | Yong et al., 2018 |

Internal friction angle ($\varphi$), cohesion ($c$) and unit weight ($\gamma$) are derived from Geological Engineering
Handbook (Geological Engineering Handbook Editorial Committee, 2018)


**Table 2**
Station records included three components of the peak ground acceleration.

| No. | Station | Epicentral distance (km) | EW (g) | NS (g) | UD (g) | Average of horizontal components (g) |
|---|---|---|---|---|---|---|
| 1 | Longtoushan 1 | 8.114 | 0.5141 | 0.9679 | 0.7193 | 0.7410 |
| 2 | Longtoushan 2 | 8.3 | 0.9685 | 0.7203 | 0.5147 | 0.8444 |
| 3 | Qianchang | 18.6 | 0.1490 | 0.1432 | 0.0539 | 0.1461 |
| 4 | Ciyuan | 32.6 | 0.0468 | 0.0457 | 0.0265 | 0.0463 |
| 5 | Mashu | 38.5 | 0.1380 | 0.1361 | 0.0663 | 0.1370 |
| 6 | Qiaojia | 43 | 0.0253 | 0.0210 | 0.0135 | 0.0232 |
| 7 | Zhaotong 1 | 47.4 | 0.0096 | 0.0152 | 0.0065 | 0.0124 |
| 8 | Zhaotong 2 | 47.671 | 0.0065 | 0.0096 | 0.0088 | 0.0081 |
| 9 | Huidongxijie | 63.3 | 0.0123 | 0.0128 | 0.0037 | 0.0126 |
| 10 | Maolin | 64.4 | 0.0251 | 0.0184 | 0.0111 | 0.0217 |
| 11 | Yongshanmaolin | 65.647 | 0.0111 | 0.0252 | 0.0184 | 0.0182 |
| 12 | Jingan | 66.2 | 0.0103 | 0.0122 | 0.0062 | 0.0113 |
| 13 | Butuotuojue | 66.8 | 0.0118 | 0.0173 | 0.0079 | 0.0146 |
| 14 | Zhaotongjingan | 67.392 | 0.0062 | 0.0103 | 0.0122 | 0.0083 |
| 15 | Huidongqianxin | 67.4 | 0.0224 | 0.0223 | 0.0067 | 0.0224 |
| 16 | Ningnansongxin | 69.2 | 0.0062 | 0.0081 | 0.0032 | 0.0071 |
| 17 | Pugebaishui | 76 | 0.0152 | 0.0149 | 0.0066 | 0.0151 |
| 18 | Huize | 76.5 | 0.0164 | 0.0182 | 0.0090 | 0.0173 |
| 19 | Pugediban | 81.2 | 0.0186 | 0.0127 | 0.0046 | 0.0156 |
| 20 | Butuodiban | 83.7 | 0.0024 | 0.0021 | 0.0024 | 0.0023 |
| 21 | Tuobuka | 85.2 | 0.0168 | 0.0168 | 0.0136 | 0.0168 |





| 22 | Pugeyangwo | 91.4 | 0.0066 | 0.0069 | 0.0022 | 0.0068 |
| 23 | Daguan | 91.8 | 0.0043 | 0.0035 | 0.0027 | 0.0039 |
