# Peer review of "An improved method of Newmark analysis for mapping hazards of coseismic landslides Mingdong Zang1,2,3, Shengwen Qi1,2,3, Yu Zou1,2,3, Zhuping Sheng4, Blanca S. Zamora4 1Key Laboratory of Shale Gas and Geoengineering, Institute of Geology and Geophysics, Chinese Academy of Sciences, Beijing 100029, China 2Institutions of Earth Science, Chinese Academy of Sciences, Beijing 100029, China 3University "

_Natural Hazards and Earth System Sciences, 2019_

## Referee Comment (RC1) · Anonymous Referee #1 · 11 Nov 2019

This study applies an approach for mapping slope susceptibility to seismic failures, which, within the framework of the standard Newmark analysis, introduces some elements of novelty in the evaluation of critical acceleration and of hazard representation. The topic can be certainly of interest for NHESS readers and, in my opinion, requires a minor-moderate revision to reach a shape acceptable for publication. Main comments are listed below.

1) A general effort of editing should be carried out to clarify some passages and improve the readability of the manuscript. On the enclosed copy, I marked statements that need to be corrected and/or rephrased, reporting some suggestions, for which, however, the authors should verify that correctly reflect what they meant.

2) The geological setting of the study area seems to be too poorly illustrated: an at

least schematic map of the study area geology should be provided.

3) With regard to the statements of lines 156-159, it is unclear to me why an angle of 45° + 1/2 of friction angle was assumed as representative of sliding surface inclination on slopes where DEM provides angle greater than 60°. Clarification would be desirable.

4) At lines 197-199, the authors declare that the critical acceleration was found to reach, in the study area, a maximum of 14 g in areas of lower susceptibility. This maximum seems to me meaningless for the context of a dynamic slope stability analysis. It could be reported that the areas with lower susceptibility are those with critical acceleration greater than 1 g.

5) At lines 262-263, the authors state that "most of the actual triggered landslides lie in the higher confidence-level areas with CF values greater than 0.60". It would be desirable to have a more quantitative information at this regard: what is the percentage of such landslides?

6) In my opinion, Fig. 15 is poorly significant. The certainty factor CF and the proportion p(H/E) occupied by landslides within areas falling in Newmark Displacement bins are two quantities uniquely related to each other, once the "a priori" probability p(H) is fixed, through the equations (9) and the corresponding inverse functions reported on Fig. 15. Thus, the perfect fitting of black dots along the red curve depends merely by the fact that the black dots are randomly selected samples of the red curve itself. More significant would be to show how CF values are related to the Newmark Displacement values Dn, as Jibson et al. (2000) did for the proportion of landslide cells, corresponding to what is here defined p(H/E), plotted versus Dn. In that study, it was this relation that was modelled through a Weibull curve, whose coefficients were derived by regression (see Fig. 14 of the cited paper). A similar plotting of CF as function of Dn would make possible to evaluate the consistency between these two quantities. Thus, one could obtain hazard estimates also for a seismic scenario different from the one used in the present study, once, following the same procedure described here, the Dn values expected for the new scenario is calculated.

Please also note the supplement to this comment:
https://www.nat-hazards-earth-syst-sci-discuss.net/nhess-2019-274/nhess-2019-274-RC1-supplement.pdf

**Supplement:**

[revised manuscript text omitted]

---

## Referee Comment (RC2) · Anonymous Referee #2 · 8 Dec 2019

In view of the problems existing in the conventional Newmark method, this paper considers the roughness and size effect of the potential slide surface unloading joint, which has some improvement and innovation on the Newmark method. But there are a minor-moderate revision before acceptance. (1) The English writing should be improved. We suggest to be polished by a native English speaker. (2) The geological map of Ludian earthquake should be added in Fig.1 including tectonic setting and the distribution of the faults. (3) About introducing the CF method, I suggest to fit the Weibull curve as Jibson et al. (2000).
* * *

---

## Author Comment (AC1) · 27 Dec 2019

Thank you very much for your kind and constructive suggestions in such detail. And we deeply appreciate your fairly good patience, the time and energy devoted for reviewing our manuscript. We have modified the manuscript following your kind comments one by one.

Q1: A general effort of editing should be carried out to clarify some passages and improve the readability of the manuscript. On the enclosed copy, I marked statements that need to be corrected and/or rephrased, reporting some suggestions, for which, however, the authors should verify that correctly reflect what they meant. R1: We have modified the manuscript following your kind comments one by one, see Line 19, 44,

50, 78-80, 82-85, 142-144, 149-150, 152, 157-159, 161, 170, 172, 174, 181, 185-188, 192-193, 196-197, 199, 204, 206, 218, 221, 223, 254, 259, 265-266, 278, 280-298, 302, 308, 310-319, 323, 326, 334, 521-522, 541-542 and 574-577 in the revision.

Q2: The geological setting of the study area seems to be too poorly illustrated: an at least schematic map of the study area geology should be provided. R2: Yes, we have added a geologic map of the study area showing lithology and faults, see Line 83 in the revision.

Q3: With regard to the statements of lines 156-159, it is unclear to me why an angle of 45° + 1/2 of friction angle was assumed as representative of sliding surface inclination on slopes where DEM provides angle greater than 60°. Clarification would be desirable. R3: According to Jibson et al. (1998, 2000), slopes steeper than 60° remained unstable, even at rather high strengths. We think that this is because Newmark's rigid-plastic block is not suitable for such steep sliding surface. In this case, the sliding would occur along a plane with an angle of 45° + 1/2 of friction angle to the horizon. We have added a Figure to make it clear, see Line 159 in the revision.

Q4: At lines 197-199, the authors declare that the critical acceleration was found to reach, in the study area, a maximum of 14 g in areas of lower susceptibility. This maximum seems to me meaningless for the context of a dynamic slope stability analysis. It could be reported that the areas with lower susceptibility are those with critical acceleration greater than 1 g. R4: Changes were made in the revision, see Line 196 in the revision.

Q5: At lines 262-263, the authors state that "most of the actual triggered landslides lie in the higher confidence-level areas with CF values greater than 0.60". It would be desirable to have a more quantitative information at this regard: what is the percentage of such landslides? R5: The quantitative portion of the actual triggered landslides lie in the higher confidence-level areas with CF values greater than 0.60 is 73.2%. Changes were made in the revision, see Line 259 in the revision.

Q6: In my opinion, Fig. 15 is poorly significant. The certainty factor CF and the proportion p(H/E) occupied by landslides within areas falling in Newmark Displacement bins are two quantities uniquely related to each other, once the "a priori" probability p(H) is fixed, through the equations (9) and the corresponding inverse functions reported on Fig. 15. Thus, the perfect fitting of black dots along the red curve depends merely by the fact that the black dots are randomly selected samples of the red curve itself. More significant would be to show how CF values are related to the Newmark Displacement values Dn, as Jibson et al. (2000) did for the proportion of landslide cells, corresponding to what is here defined p(H/E), plotted versus Dn. In that study, it was this relation that was modelled through a Weibull curve, whose coefficients were derived by regression (see Fig. 14 of the cited paper). A similar plotting of CF as function of Dn would make possible to evaluate the consistency between these two quantities. Thus, one could obtain hazard estimates also for a seismic scenario different from the one used in the present study, once, following the same procedure described here, the Dn values expected for the new scenario is calculated. R6: We consider your suggestion seriously and modify the manuscript depicted as follows: (1) for the statistical significance of the function between CF and Newmark displacement. The predicted displacement cells were grouped into bins based on quantile statistics. The break points are 0, 10, 30, 39, 46, 51, 55, 59, 63 and 122 respectively. In this way, the number of cells in each bin is equal; (2) As the range of CF-values is from -1 to 1, not 0 to 1, Weibull (1939) curve developed by Jaeger and Cook (1969) is unsuitable here. Therefore, we modified the functional form to CF=2m[1-exp(-aD^b)]^-1, where CF is the certainty factor, m is the maximum CF-value represented by the data, D is predicated displacement, and a and b are regression constants. In each bin, the CF-value of Newmark displacement was plotted as a dot. The regression curve based on data from the Ludian earthquake is CF=1.837[1-exp(-0.073D^0.821)]^-1; (3) The value of CF increase monotonically with increasing Newmark displacement. We could obtain hazard estimates different from the one used in the present study through following the same procedure described here. Changes were made in the revision, see Line

280-298.

---

## Author Comment (AC2) · 27 Dec 2019

Thanks very much for your nice comment. And we deeply appreciate your time devoted by reviewing our manuscript. Your constructive comments are invaluable to the improvement of our manuscript.

Q1: The English writing should be improved. We suggest to be polished by a native English speaker. R1: Yes, we have polished the manuscript with the help of a native English speaker.

Q2: The geological map of Ludian earthquake should be added in Fig.1 including tectonic setting and the distribution of the faults. R2: Yes, we have added a geologic map of the study area showing the distribution of lithology and faults, see Line 83 in

the revision.

Q3: About introducing the CF method, I suggest to fit the Weibull curve as Jibson et al. (2000). R3: Yes, we have modified Weibull function form developed by Jaeger and Cook (1969) to CF=2m[1-exp(-aD^b)]^-1, where CF is the certainty factor, m is the maximum CF-value represented by the data, D is predicated displacement, and a and b are regression constants. The CF-value of Newmark displacement was plotted as a dot. The regression curve based on data from the Ludian earthquake is CF=1.837[1-exp(-0.073D^0.821)]^-1. With this function, we could obtain hazard estimates different from the one used in the present study through following the same procedure described here. Changes were made in the revision, see Line 280-298.

---

## Author Response (AR1)

**Responses to Referee #1**

Thank you very much for your kind and constructive suggestions in such detail. And we deeply appreciate your fairly good patience, the time and energy devoted for reviewing our manuscript. We have modified the manuscript following your kind comments one by one.

**Q1:** A general effort of editing should be carried out to clarify some passages and improve the readability of the manuscript. On the enclosed copy, I marked statements that need to be corrected and/or rephrased, reporting some suggestions, for which, however, the authors should verify that correctly reflect what they meant.

**R1:** We have modified the manuscript following your kind comments one by one, see Line 16, 43, 48, 79-84, 85, 141-145, 149-150, 155-159, 169, 181, 185-188, 193-194, 195-197, 204-206, 218, 221, 254, 266, 278, 302, 309-310, 310-312, 312-315, 315-316, 464, 486, 510 and 538 in the revision.

**Q2:** The geological setting of the study area seems to be too poorly illustrated: an at least schematic map of the study area geology should be provided.

**R2:** Yes, we have added a geologic map of the study area showing lithology and faults, see Line 84 and 541 in the revision.

**Q3:** With regard to the statements of lines 156-159, it is unclear to me why an angle of 45° + 1/2 of friction angle was assumed as representative of sliding surface inclination on slopes where DEM provides angle greater than 60°. Clarification would be desirable.

**R3:** According to Jibson et al. (1998, 2000), slopes steeper than 60° remain unstable even at high strengths. We assume that Newmark's rigid plastic block is unsuitable for such a steep sliding surface. In this case, sliding occurs along a plane at an angle ($\alpha$) of $45°+\frac{\phi_b}{2}$ of the friction angle with the horizon. Therefore, we assigned an angle ($\alpha$) of $45°+\frac{\phi_b}{2}$ to slopes steeper than 60° to avoid too samall a value of $F_S$ in the Newmark analysis. We have added a figure to make it clear, see Line 155-159 and 561 in the revision.

**Q4:** At lines 197-199, the authors declare that the critical acceleration was found to reach, in the study area, a maximum of 14 g in areas of lower susceptibility. This maximum seems to me meaningless for the context of a dynamic slope stability analysis. It could be reported that the areas with lower susceptibility are those with critical acceleration greater than 1 g.

**R4:** Changes were made in the revision, see Line 195-197 in the revision.

**Q5:** At lines 262-263, the authors state that "most of the actual triggered landslides lie in the higher confidence-level areas with CF values greater than 0.60". It would be desirable to have a more quantitative information at this regard: what is the percentage of such landslides?

**R5:** The quantitative portion of the actual triggered landslides lie in the higher confidence-level areas with CF values greater than 0.60 is 73.2%. Changes were made in the revision, see Line 259-260 in the revision.

**Q6:** In my opinion, Fig. 15 is poorly significant. The certainty factor CF and the proportion p(H/E) occupied by landslides within areas falling in Newmark Displacement bins are two quantities uniquely related to each other, once the "a priori" probability p(H) is fixed, through the equations (9) and the corresponding inverse functions reported on Fig. 15. Thus, the perfect fitting of black dots along the red curve depends merely by the fact that the black dots are randomly selected samples of the red curve itself. More significant would be to show how CF values are related to the Newmark Displacement values Dn, as Jibson et al. (2000) did for the proportion of landslide cells, corresponding to what is here defined p(H/E), plotted versus Dn. In that study, it was this relation that was modelled through a Weibull curve, whose coefficients were derived by regression (see Fig. 14 of the cited paper). A similar plotting of CF as function of Dn would make possible to evaluate the consistency between these two quantities. Thus, one could obtain hazard estimates also for a seismic scenario different from the one used in the present study, once, following the same procedure described here, the Dn values expected for the new scenario is calculated.

**R6:** We consider your suggestion seriously and modify the manuscript depicted as follows: (1) for the statistical significance of the function of CF and Newmark displacement., the predicted displacement cells were grouped into bins based on quantile statistics. The breakpoints were 0, 10, 30, 39, 46, 51, 55, 59, 63, and 122. In this way, the number of cells in each bin was equal.; (2) as CF values ranged from -1 to 1, and not from 0 to 1, the Weibull (1939) curve developed by Jaeger and Cook (1969) is unsuitable here. Therefore, we modified the functional form to $CF=2m[1-\exp(-aD^b)]-1$, where CF is the certainty factor, m is the maximum CF value represented by the data, D is predicated displacement, and a and b are regression constants. In each bin, the CF value of Newmark displacement was plotted as a dot. The regression curve based on data from the Ludian earthquake is $CF=1.837[1-\exp(-0.073D^{0.821})]-1$; (3) when the predicted displacement increased, the value of CF increased monotonically, meaning that the confidence level for slope failure grew and landslide would probably occur. Such a procedure is consistent with the interpretation of certainty factor theory. Therefore, we were able to obtain estimates of the hazard different from the one used in this study using the same procedure described here. Changes were made in the revision, see Line 280-298.

**Responses to Referee #2**

Thanks very much for your nice comment. And we deeply appreciate your time devoted by reviewing our manuscript. Your constructive comments are invaluable to the improvement of our manuscript. We have modified the manuscript following your comments one by one.

**Q1:** The English writing should be improved. We suggest to be polished by a native English speaker.

**R1:** Yes, we have polished the manuscript with the help of a native English speaker, changes were made in the revision.

**Q2:** The geological map of Ludian earthquake should be added in Fig.1 including tectonic setting and the distribution of the faults.

**R2:** Yes, we have added a geologic map of the study area showing the distribution of lithology and faults, see Line 84 and 541 in the revision.

**Q3:** About introducing the CF method, I suggest to fit the Weibull curve as Jibson et al. (2000).

**R3:** Yes, we modified the Weibull function form developed by Jaeger and Cook (1969) to $CF=2m[1-\exp(-aD^b)]-1$, 
[revised manuscript text omitted]

| Limestone | 21.5 | 37° | 160 | 9 | 45° | 30 | Bandis et al., 1983
 Singh et al., 2012
 Yong et al., 2018 |
| Shale | 24.9 | 27° | 75 | 8 | 27° | 16 | Barton and Choubey, 1977
 Bilgin and Pasamehmetoglu, 1990 |
| Sandstone | 23.5 | 35° | 100 | 6 | 42° | 24 | Coulson, 1972
 Bandis et al., 1983
 Priest, 1993 |
| Basalt | 27.9 | 38° | 205 | 8.5 | 50° | 40 | Coulson, 1972
 Barton and Choubey, 1977
 Alejano et al., 2014 |
| Slate | 26.5 | 30° | 175 | 3 | 40° | 11 | Coulson, 1972
 Barton and Choubey, 1977
 Bandis et al., 1983
 Alejano et al., 2012
 Yong et al., 2018 |

Friction angle ($\varphi$), cohesion ($c$), and unit weight ($\gamma$) were derived from the Geological Engineering

Handbook (Geological Engineering Handbook Editorial Committee, 2018)

**Table 2**

Station records of three components of peak ground acceleration.

| No. | Station | Epicentral distance (km) | EW (g) | NS (g) | UD (g) | Average of horizontal components (g) |
|---|---|---|---|---|---|---|
| 1 | Longtoushan 1 | 8.114 | 0.5141 | 0.9679 | 0.7193 | 0.7410 |
| 2 | Longtoushan 2 | 8.3 | 0.9685 | 0.7203 | 0.5147 | 0.8444 |
| 3 | Qianchang | 18.6 | 0.1490 | 0.1432 | 0.0539 | 0.1461 |
| 4 | Ciyuan | 32.6 | 0.0468 | 0.0457 | 0.0265 | 0.0463 |
| 5 | Mashu | 38.5 | 0.1380 | 0.1361 | 0.0663 | 0.1370 |
| 6 | Qiaojia | 43 | 0.0253 | 0.0210 | 0.0135 | 0.0232 |
| 7 | Zhaotong 1 | 47.4 | 0.0096 | 0.0152 | 0.0065 | 0.0124 |
| 8 | Zhaotong 2 | 47.671 | 0.0065 | 0.0096 | 0.0088 | 0.0081 |
| 9 | Huidongxijie | 63.3 | 0.0123 | 0.0128 | 0.0037 | 0.0126 |
| 10 | Maolin | 64.4 | 0.0251 | 0.0184 | 0.0111 | 0.0217 |
| 11 | Yongshanmaolin | 65.647 | 0.0111 | 0.0252 | 0.0184 | 0.0182 |
| 12 | Jingan | 66.2 | 0.0103 | 0.0122 | 0.0062 | 0.0113 |
| 13 | Butuotuojue | 66.8 | 0.0118 | 0.0173 | 0.0079 | 0.0146 |
| 14 | Zhaotongjingan | 67.392 | 0.0062 | 0.0103 | 0.0122 | 0.0083 |
| 15 | Huidongqianxin | 67.4 | 0.0224 | 0.0223 | 0.0067 | 0.0224 |
| 16 | Ningnansongxin | 69.2 | 0.0062 | 0.0081 | 0.0032 | 0.0071 |
| 17 | Pugebaishui | 76 | 0.0152 | 0.0149 | 0.0066 | 0.0151 |
| 18 | Huize | 76.5 | 0.0164 | 0.0182 | 0.0090 | 0.0173 |
| 19 | Pugediban | 81.2 | 0.0186 | 0.0127 | 0.0046 | 0.0156 |
| 20 | Butuodiban | 83.7 | 0.0024 | 0.0021 | 0.0024 | 0.0023 |
| 21 | Tuobuka | 85.2 | 0.0168 | 0.0168 | 0.0136 | 0.0168 |

| 22 | Pugeyangwo | 91.4 | 0.0066 | 0.0069 | 0.0022 | 0.0068 |
| 23 | Daguan | 91.8 | 0.0043 | 0.0035 | 0.0027 | 0.0039 |